# *Azorella compacta* Organic Extracts Exacerbate Metabolic Dysfunction-Associated Fatty Liver Disease in Mice Fed a High-Fat Diet

**DOI:** 10.3390/ph17060746

**Published:** 2024-06-06

**Authors:** Jessica Zúñiga-Hernandez, Matías Quiñones San Martin, Benjamín Figueroa, Ulises Novoa, Francisco A. Monsalve, Mitchell Bacho, Aurelio San-Martin, Daniel R. González

**Affiliations:** 1Department of Basic Biomedical Sciences, Faculty of Health Sciences, Universidad de Talca, Talca 3460000, Chile; jezuniga@utalca.cl (J.Z.-H.);; 2Doctorado en Ciencias, Mención I+D de Productos Bioactivos, Instituto de Química de Recursos Naturales, Universidad de Talca, Talca 3341717, Chile; 3Department of Preclinical Sciences, Faculty of Medicine, Universidad Católica del Maule, Talca 3466706, Chile; fmonsalve@ucm.cl; 4Departamento de Ciencias Químicas, Facultad de Ciencias Exactas, Laboratorio de Síntesis Orgánica y Organometálica, Universidad Andrés Bello, Santiago 8370146, Chile; 5Departamento de Ciencias y Recursos Naturales, Facultad de Ciencias, Universidad de Magallanes, Punta Arenas 6200112, Chile

**Keywords:** *Azorella compacta*, metabolic syndrome, diterpenoids, mulinane, azorellane, non-alcoholic fatty liver disease

## Abstract

*Azorella compacta* (*A. compacta*) is a shrub of the Andean Altiplano of Bolivia, Chile and Peru, consumed by local communities as a traditional medicine for several maladies such as diabetes, hepatic and inflammatory diseases. *A. compacta* is rich in mulinane- and azorellane-type diterpenoids. For two of these, acute hypoglycemic effects have been described, but the impact of *A. compacta* diterpenoids on fatty liver disease has not been investigated. Therefore, *A. compacta* organic fractions were prepared using petroleum ether, dichloromethane and methanol. Their content was characterized by UHPLC/MS, revealing the presence of ten diterpenoids, mainly mulinic acid, azorellanol and mulin-11,13-diene. Next, mice fed with a high-fat diet (HFD), a model of metabolic dysfunction-associated fatty liver disease (MAFLD), received one of the fractions in drinking water for two weeks. After this treatment, hepatic parameters were evaluated. The *A. compacta* fractions did not reduce hyperglycemia or body weight in the HFD-fed mice but increased the serum levels of hepatic transaminases (AST and ALT), reduced albumin and increased bilirubin, indicating hepatic damage, while histopathological alterations such as steatosis, inflammation and necrosis generated by the HFD were, overall, not ameliorated by the fractions. These results suggest that organic *A. compacta* extracts may generate hepatic complications in patients with MAFLD.

## 1. Introduction

*Azorella compacta* (Phil. *Apiaceae*) is a native shrub that grows in the Andean region of the north part of Chile, south Peru, Bolivia and north Argentina at high altitudes (above 4000 mt above sea level) [1]. It is a cushion plant which presents green woody mounds, and it is able to survive in geographical conditions where very few other plants grow [2]. Known by the locals as “yareta”, *Azorella compacta* (*A. compacta*) is used in this part of the Andean Altiplano (highlands) as a medicinal plant for the treatment of inflammatory diseases, diabetes and renal and hepatic conditions, among others [3]. It is consumed as an infusion made from its roots and resins for topical use [4]. Currently, *A. compacta* is sold as a herbal medicine over the rest of the continent as a dry plant.

*A. compacta* is rich in terpenoids, particularly diterpenoids of the types mulinane and azorellane [5,6]. Mulinanes present a tricyclic skeleton of 21 carbon atoms [6,7,8], while azorellanes have a tetracyclic structure of 20 carbon atoms [9,10]. To date, 27 diterpenoids from *A. compacta* have been isolated and identified: 11 mulinanes and 6 azorellanes [10].

Several biological activities have been attributed to these *A. compacta*-derived secondary metabolites. For instance, mulinanes and azorellanes exhibit antimicrobial and cytotoxic activity [10]. At a systemic level, molecules that belong to these families of terpenoids have been reported to show anti-ulcer and anti-inflammatory activities, consistent with the use of *A. compacta* in folk medicine [10]. Interestingly, mulinolic acid and azorellanol isolated from *A. compacta* presented anti-diabetic activity when evaluated acutely in rats [11,12]. Nevertheless, neither these compounds nor *A. compacta* extracts have been tested as a chronic treatment.

Non-alcoholic fatty liver disease (NAFLD) is a complex syndrome characterized by triglyceride accumulation in hepatocytes (steatosis), developed in the absence of excessive alcohol consumption [13]. NAFLD is the hepatic manifestation of metabolic syndrome, which includes hyperglycemia, obesity, hypertension and dyslipidemia [14]. Hence, it is usually observed in a range of conditions that include overweight and obesity. Recently, an international panel of experts modified the NAFLD nomenclature to “metabolic dysfunction-associated fatty liver disease” (MAFLD), incorporating a set of medical/biochemical criteria to enable an easier diagnosis [15]. The main criteria for NAFLD/MALFLD are a hepatic steatosis of ≥5% hepatocytes without current liver disease and two metabolic risks (metabolic deregulatory factors) such as increased waist circumference, high blood pressure, increased plasma triglycerides, HDL and insulin resistance, among others [16]. On the other hand, type 2 diabetes mellitus (T2DM) and/or overweight/obesity are classically involved in liver fat deposition and have been associated with disease progression and hepatic and extra-hepatic complications [15]. MAFLD affects approximately 30% of adults in the general population and up to 70% of patients with T2DM [17]. It includes a spectrum of progressive conditions, such as simple steatosis (a benign and reversible liver disorder), steatohepatitis (NASH), varied severity degrees of fibrosis and eventually cirrhosis and hepatocellular carcinoma [13,18].

Currently, MAFLD is a condition that has no established treatment. The most suitable approaches to reducing steatosis include lifestyle changes and drug therapy as a second strategy [19]. Treatments such as lifestyle modifications and body weight loss associated with bariatric surgery show modest benefits, and the adverse effects of the treatments complicate their use [14]. There are studies based on natural supplements and derivatives indicating that chronic liver disease complications could be reversed and improved by these products [20,21,22,23]. Interestingly, terpenoids, which are abundant in *A. compacta*, have been proposed for the treatment of NAFLD [22]. Therefore, here, we aimed to evaluate the impact of the administration of *A. compacta* in a chronic manner to mice fed a high-fat diet as a model of metabolic disease.

## 2. Results

### 2.1. Metabolomic Characterization of Azorella compacta Extracts

Methanol, dichloromethane and petroleum ether fractions of *A. compacta* were characterized. For this, aliquots of each fraction were subjected to Ultrahigh-Performance Liquid Chromatography (UHPLC)–mass spectrometry identification. With this methodology, a total of ten diterpenoids were identified (Table 1 and Figure 1). In addition, four different secondary metabolites were identified: 5.7-dihydroxychromone (**11**), biochanin A (**12**), azelaic acid (**13**) and homoorientin (**14**).

The structures of the diterpenoids identified are shown in Figure 1. Of these, compounds **1**, **2a** and **2b** correspond to azorellanes, and compounds **3**–**10** are mulinane-type diterpenoids. The relative abundance of each fraction is shown in Figure 2. In the petroleum ether fraction, mulinic acid (**3**) and 11,12-epoxymulin-13-en-20-oic acid (**5**) dominated. The content of the dichloromethane fraction was more heterogeneous (nine compounds). It had high concentrations of azorellanes (**1**–**3**), mainly α and β azorellanol, but also included a significant amount of mulinic acid (23%), while the methanol fraction (the most polar) was also dominated by mulinic acid (57%), but also contained 30% mulin 11,13 diene (**10**).

### 2.2. Impact of Azorella compacta Organic Fractions on Mice Fed a High-Fat Diet

Next, the impact of the three organic extracts from *A. compacta* was analyzed in a model of metabolic syndrome since *A. compacta* compounds have been described to reduce hyperglycemia in diabetic rats. For this purpose, mice were fed a high-fat diet for eight weeks. Then, hyperglycemic mice received one of the *A. compacta* fractions, HFD Az-met (methanol extract), HFD Az-DCM (dichloromethane extract), and HFD Az-ether, (petroleum ether extract) or a vehicle in drinking water for two weeks while maintaining the high-fat diet regime. During this period, none of the animals that received the extracts died. As shown in Figure 3, at the end of the protocol, the HFD-fed mice presented an increase in body weight (Figure 3A) and blood glucose (Figure 3B). The HFD Az-DCM- and HFD Az-met-treated mice did not present changes in body weight compared to the HFD control animals, while the Az-ether group had increased body weight by 17% compared to the HFD control (36.3 g to 42.7 g, *p* = 0.0132). A similar situation was observed for the analysis of glycemia (Figure 3B). None of the extracts reduced the hyperglycemia observed in the HFD mice: 187.4 ± 36.1, 233.5 ± 56.8, 179.6 ± 23.9 and 209.5 ± 23.5 mg/dL for the HFD-, Az-met-, Az-DCM- and Az-ether-treated groups, respectively. Notably, the mice that received Az-met presented an increase in blood glucose of 93% and 24% compared to the animals that received the normal and HFD diets, respectively.

Next, hepatic biochemical parameters were analyzed in the serum of the treated animals (Figure 4). The enzymatic activity of hepatic transaminases ALT and AST was augmented in the serum of the animals that received any of the fractions, with respect to the control and HFD-alone-treated mice (Figure 4A,B). Regarding other parameters of liver function, albumin was decreased in the Az-ether group by 55 and 44.5% with respect to the control and HFD alone groups, respectively (Figure 4C). Serum creatine kinase (CK) did not present statistical differences among the groups, and all levels remained within normal laboratory ranges, suggesting no impact on the heart or skeletal muscle (Figure 4D). Total bilirubin levels presented significant increases in the serum of all the groups treated with the *A. compacta* organic fractions (Figure 4E). The averages observed for total bilirubin (µM) were 2.5 ± 0.9, 6.5 ± 5.4, 4.3 ± 2.8, 20.5 ± 1.6 and 12.3 ± 9.3 for the control, HFD, Az-met, Az-DCM and Az-ether groups, respectively. The highest levels were observed in the Az-DCM and Az-ether groups, with an enhancement of 4.7 and 2.8 times with respect to the HFD alone group.

### 2.3. Effect of A. compacta Organic Extract Treatment on Hepatic Pathology

Hepatic damage in the HFD-treated animals and the impact of the treatment with the *A. compacta* organic fractions were evaluated through a histopathology analysis of liver sections (Figure 5). Hematoxylin/eosin-stained sections from control mice showed normal liver histoarchitecture, with the absence of necrotic foci, inflammatory infiltrates and lipid drops/steatosis, whereas the HFD mice’s liver sections displayed deterioration in these four parameters with appreciable degenerative changes, especially liver steatosis (more than 30% lipid droplet accumulation) compared to the normal diet-treated animals. The *A. compacta* extract-treated groups showed variable degrees of liver damage, which is apparent in the analysis of cytoarchitecture, inflammation, necrosis and steatosis (Figure 5B–E). Regarding hepatic cytoarchitecture damage (4B), the Az-DCM- and Az-ether-treated groups showed a 37.5 and 46% decrease compared to the HFD group (*p* = 0.0009 and 0.007, respectively), with no difference observed for the Az-met group. In the case of the necrotic foci analysis (5C), none of the groups showed differences in damage compared to the HFD group. When inflammatory cell infiltration was evaluated, the Az-ether group presented an increase in inflammatory foci (149% increase, *p* = 0.0035), while the other groups showed no differences compared to the HFD-treated mice. For steatosis, the Az-met group presented a 44% decrease (*p* = 0.002) and the Az-DCM-treated mice a 29% reduction (*p* = 0.0142) compared to the HFD alone group, while the Az-ether group presented no changes in this parameter.

In summary, the impact of the organic fractions of *A. compacta* on the pathology parameters evaluated is heterogeneous, with both positive and negative effects. Overall, a mild positive effect was observed with the Az-DCM fraction, but a slight worsening one was observed with the Az-ether fraction.

### 2.4. In Silico Analysis of Toxicity and Metabolism of A. compacta Diterpenoids

Given the potential for hepatic damage under the conditions studied, we used an in silico analysis for the toxicological and metabolic parameters of the ten diterpenoids found in the organic fractions using the ADMET 2.0 platform (Table 2). For this, we first focused on parameters with the potential to induce hepatotoxicity in humans (H-HT) and drug-induced liver injury (DILI). For these parameters, compound **4** exhibited an intermediate risk for HILI, while compound **9** presented a high probability of inducing DILI. To evaluate parameters of biotransformation, we focused on the potential of the terpenoids as being substrates or inhibitors of the CYPA2/2C19/2C9/2D6/3A4 isoforms of the human cytochrome p450 family, which produce phase I (oxidative) reactions mainly in the liver. In general, all the compounds analyzed presented a very high potential to be substrates of CYPAC19; half of them (to a lesser extent) had the potential to be substrates and/or inhibitors of CYP3A4; and almost none were likely to be substrates or inhibitors of CYP2D6. They also presented, in general, a significant probability (although to a lesser extent) of being substrates of CYP3A4. None of the compounds presented significant probabilities of being inhibitors of CYP2C9. In addition, several potential toxicological targets were also analyzed in silico. Of these, the most prominent profiles found included a stress response (SR) to activate p53 and affect the mitochondrial membrane potential (MMP) in eight and nine of the compounds studied, respectively. P53 is a tumor suppressor activated by cell and DNA damage, which, under cell injury, induces cell cycle arrest, apoptosis or cellular senescence. Nine of the compounds presented a high probability of affecting MMP, and eight showed potential as activating molecules of p53, most notably compounds **2a** and **2b** (β and α azorellanol).

## 3. Discussion

*A. compacta* is used as a herbal medicine in Andes altiplano communities, mainly as infusions to treat a variety of maladies such as diabetes and inflammatory conditions [3]. Several diterpenoids have been isolated from this plant and tested for pharmacological properties, such as microbicide, anti-hyperglycemic, gastro-protective, anti-inflammatory and analgesic activities [10], but the impact of *A. compacta* in a chronic model in vivo has not been fully assessed. Here, we evaluated the effect of organic extracts from *A. compacta* in a model of metabolic disease that presents hyperglycemia, obesity and hepatic alterations. These extracts (petroleum ether, dichloromethane and methanol) contained ten diterpenoids distributed differentially among the fractions. Interestingly, the petroleum ether and methanol fractions were mainly rich in mulinic acid, whereas the dichloromethane fraction included azorellanol, mulinic acid and 13α,14α-dihydroxymulin-11-en-20-oic acid as its main components. The treatment consisted of a two-week administration of a high dose of the extracts (~50 mg per day). Notably, the treatment with these extracts did not change the hyperglycemic state or increase weight due to the HFD diet, suggesting that *A. compacta* organic fractions are not able to reduce increased glycemic levels in a model more closely resembling T2DM. Previous results in a model of T1D (rats treated with streptozotocin) showed an acute reduction in blood glucose induced by mulinolic acid and azorellanol administration but not by mulin-11,13-dien-20-oic acid [11]. Interestingly, our fractions did not contain mulinolic acid, and azorellanol was abundant mainly in the DCM fraction, but as a racemic mixture. These conditions might explain the lack of effect on hyperglycemia of the organic fractions. Importantly, in the mentioned study, rats treated with streptozotocin reached glycemia values of ~500 mg/dL. Under those conditions, azorellanol acted as an insulin secretagogue, a mechanism that might not be relevant in the present HFD model, which essentially generates insulin resistance, with blood sugar levels of approximately 200 mg/dL.

Furthermore, when we evaluated hepatic damage parameters, an exacerbation of the alterations generated by the HFD was observed. This was evident mainly in the serum levels of the transaminases AST and ALT, which were increased in the animals treated with any of the three organic fractions. Furthermore, the ether and DCM fractions reduced plasma albumin and increased bilirubin levels, both clear signs of hepatic damage.

The histopathology analyses indicated that, in general, the fractions failed to normalize the parameters of liver damage, especially necrotic foci, which correlates with the increased activity of ALT and AST found in the serum of the treated animals. Although, in terms of liver cytoarchitecture, the DCM and petroleum ether extracts reduced the damage observed, the petroleum ether fraction enhanced the inflammatory infiltrates in liver tissue. Hepatic inflammatory infiltration is a cardinal sign of liver disease [20].

The *A. compacta* organic fractions used in this study contained ten diterpenoids. Some diterpenoids have been reported to be hepatoprotective by exerting antioxidant, anti-inflammatory, hypolipidemic and anti-fibrotic effects [24]. For example, it was recently reported that ruebellapenes extracted from *Callicarpa rubella* decreased the levels of transaminases in an in vitro model of BRL/hepatocyte damage [25]. Another study showed that carnosic acid, a diterpenoid derived from rosemary leaves (*Rosmarinus officinalis*), attenuated obesity, insulin levels and liver adiposity and modulated genes associated with lipid metabolism in *ob*/*ob* mice [26]. The same positive results were found with Ginkgolide A and B in HFD-induced NAFLD mice, where these diterpenoids derived from *Ginkgo biloba L* ameliorated hepatic steatosis through the inhibition of lipid accumulation and lipoapoptosis and exerted anti-inflammatory effects [22,27,28].

However, the use of diterpenoids is controversial due to their potential toxic effects. For example, diterpenoids from *Aconitum* presented anti-inflammatory and anti-cancer activities [29] but also presented toxic effects on the nervous and cardiovascular systems [30]. Also, diterpenoids derived from *Delphinium* and *Aconitum* species have been described as cytotoxic to HepG2 cells, an activity that could be useful in hepatocarcinogenesis [31]. In addition, tripolide, a component of *Triptergium wilfordii* Hook used in the treatment of autoimmune diseases, has been described as a hepatotoxic agent [32]. Tripolide acts on hepatic influx and efflux transporters and causes severe jaundice, drug-induced hepatitis (DILI) and abnormal liver function, which obstacles to its clinical use [33,34]. Other diterpenoids that generate hepatotoxicity in humans include Paclitaxel (isolated from the Western Yew tree) and the protein kinase inhibitors Lapatinib and germander, isolated from *Teucrium lamiaceae* [24].

Analogous to DILI, an adverse toxic drug reaction resulting in liver injury, there has been established herb-induced liver injury (HILI) [35]. The incidence of HILI has increased as the use of phytotherapy has gained popularity in Western countries, considering that the most promoted herbal products lack scientific evidence of efficacy and safety [36,37]. Terpenes have been reported as HILI-inducing products. For instance, citral, a monoterpene obtained from myrtle trees, lemons, and limes, causes dose-dependent liver necrosis [38]. The sesquiterpene seredone (from *Curcuma elata*) causes hepatocyte degeneration, centrolobulillar necrosis and immune cell infiltration, increasing ALT and total bilirubin [39]. *Dioscorea bulbifera* (used in goiters and tumors) is rich in furanoterpenoids and generates hepatotoxicity associated with their oxidation by CYP3A [40].

The use of *A. compacta* in folk medicine is mainly as an aqueous infusion of dried areal parts and roots [3,4]. Indeed, those preparations may not be abundant in diterpenoids but are rich in water-soluble antioxidants, as has been shown in a study on immune cell activation by an *Azorella* infusion. In that study, the infusion of *A. compacta* included chlorogenic acid, apigenin, isoorientin (homoorientin), orientin, dicaffeoylquinic acid, biochanin A and licoisoflavone A, but not diterpenoids [41]. Our fractions contained small amounts of 5.7-dihydroxychromone, biochanin A and homoorientin as potential sources of antioxidants. On the contrary, methanol fractions of *A. compacta* have been shown to induce apoptosis in HL60 cells, a cell line of neutrophils [42], although the composition of those extracts was not described. In MCF-7 cells, a breast cancer cell line, several *A. compacta* diterpenoids exhibited cytotoxic activity [43], including 7β-deacetylazorellanol, azorellanol and mulin-11,13-dien-20-oic acid. These terpenoids (compounds **1**, **2a**/**2b** and **4** in our study) are abundant in our fractions, which is consistent with the hepatotoxic effect observed. These results are supported by the in silico ADMET analysis, which assigned compound **4** as having an intermediate probability of inducing HILI, with mitochondrial membrane potential as a target, and compounds **1** and **2a/b** as having a high probability of altering mitochondrial membrane potential and activating p53, both converging signals for cell death.

From our results, it can be speculated that *A. compacta* preparations that include terpenoids have the potential to exacerbate hepatic damage, especially in patients with established metabolic diseases such as diabetes and obesity. Particularly, the increase in total bilirubin levels should be considered a warning for potential hepatotoxic effects. Therefore, it is necessary to further evaluate the hepatotoxicity of the *A. compacta* diterpenoids.

## 4. Materials and Methods

### 4.1. Plant Material

*Azorella compacta* Phil. Apiaceae was collected in May 2013 in the El Tatio geysers area (Atacama Desert) at an elevation of 4200 m in the Region of Antofagasta, Chile. A sample specimen was deposited at the Herbarium of Natural Products Laboratory of the Faculty of Science, Universidad de Chile, under the code n° 0513 [44]. Leaves and stems were separated and dried at ambient temperature, avoiding direct sun exposure, over three days. After this period, areal parts (127 g) were macerated and extracted, first with petroleum ether (three times), then with dichloromethane and finally with methanol. Then, the extracts were filtered and dried under reduced pressure. All chemical reagents used were of analytical grade, obtained from Merck (Merck KGaA, Darmstadt, Germany)

### 4.2. Metabolomic Analysis

An aliquot of each dried extract was prepared for the metabolomics analysis by UltraHigh-Performance Liquid Chromatography (UHPLC)–mass spectrometry (Compact QTOF MS + Elute UHPLC, Bruker Daltonik GmbH, Bremen, Germany). Petroleum ether, dichloromethane and methanol fractions were solubilized in 500 µL of hexane, dichloromethane and ethanol, respectively, and transferred to HPLC vials. The chromatographic separation of 5 μL of each extract was carried out in a Kinetex C18 (Phenomenex, Torrence, CA, USA) (2.1 mm × 100 mm, particle size 1.7 µm) at 0.5 mL/min and 35 °C using 0.1% formic acid in water (A) and 0.1% formic acid in acetonitrile (B) as mobile phases, according to the following gradient: 0 min, 85% A; 18 min, 10% A; 20 min, 10% A; 30 min, 85% A; 35 min, 85% A. The total separation time was 35 min. Mass spectrometry data were acquired over a range of *m*/*z* 120–1800 in the negative and positive ion modes of the electrospray ionization (ESI) source: sheath gas: 30 psi; auxiliary gas: 13; spray voltage: 3 kV; capillary temperature: 350 °C; S-Lens RF level: 50; heater temperature: 150 °C. Mass spectrometry was performed in positive and negative modes. Data were analyzed using Metaboscape 4.0 software (Bruker, Billerica, MA, USA), using the MassBank of North America (Mona, Davis, CA, USA) spectra library for metabolite identification.

### 4.3. Mice and Diets

Adult male C57Bl6 mice (6–8 weeks old), obtained from the animal facility of the Universidad de Talca, were used. The study protocol (N° 2015-04) was approved by the Institutional Animal Care and Use Committee (CIECUAL) of Universidad de Talca. Animals were kept at 21–25 °C in cycles of 12 h light/darkness with water and chow ad libitum. Animals were randomly selected to receive normal (Prolab rat/mouse/hamster 3000) or high-fat diets (DIO rodent purified diet, 45% energy from fat), both from LabDiet (St. Louis, MO, USA) for eight weeks. After this period, animals on the DIO diet were checked for glycemia (obtained from a tail puncture using a Hemoglucotest device). Those that presented values above 150 mg/dL were randomly selected to receive either a vehicle (*n* = 5) or one of the *A. compacta* extracts (*n* = 5 each group). The fractions (13.4 g each) were reconstituted in 2 mL of the vehicle (12% Tween 80 in NaCl 0.9%) and administrated in drinking water (final concentration: 13.4 mg/mL). Since mice drink, on average, 4–6 mL/day, each animal received approximately 50 mg/day. The animals received the extracts for two weeks in drinking water, maintaining the high-fat diet regime. At the end of the experimental period, the animals were euthanized through the administration of 3% isoflurane to induce anesthesia, then ketamine (Ketoshop, Drag Pharma Invetec S.A, Santiago, Chile) at 80 mg/kg and xylazine (Xylavet, Agroland, Santiago, Chile) at 12 mg/kg, both intraperitoneally. Under deep anesthesia, the animals were exsanguinated by cardiac puncture for plasma and organ extraction.

### 4.4. Determination of Biochemical Parameters

Serum was obtained by blood clotting for three hours at room temperature and centrifugation at 2000× *g* for 5 min. Alanine aminotransferase (ALT), aspartate aminotransferase (AST) and creatine kinase (CK) activities were measured using specific diagnostic kits (ALT, AST and Valtek^®^ Diagnostic Kits, Ñuñoa, Chile). Albumin and bilirubin were measured using kits from LiquidColor Human™ (Wiesbaden, Germany), and Stanbio Total & Direct Bilirrubin^®^ (StanBio, Boerne, TX, USA), respectively. Adequate two-level controls, normal and pathological, were used. All the serum assays were evaluated in triplicate.

### 4.5. Liver Histopathology

After extraction, the livers were immersed in Bouin solution for 12–24 h. After this period, pieces of the tissues were dehydrated in alcohol solutions and included in Paraplast. After inclusion, 5 µm sections were obtained using a microtome and mounted on 0.1% polylysine-treated coverslips. After this, sections were stained with hematoxylin/eosin in an automatized tissue processor, Leica TP1020 (Leica Microsystems Inc. (Schweiz) AG, Heerbrugg, Switzerland) and Leica EG11504 H (Leica Microsystems Inc.).

The examination was performed by two investigators in a blinded fashion. Twenty random fields were assessed for (i) necrosis, which was evaluated according to the Korourian score [45]: none = 1, occasional (1%) necrotic hepatocyte = 1, frequent (5–10%) necrotic hepatocytes = 2, small foci of necrosis (clusters greater than 10 necrotic hepatocytes) = 3, and extensive areas of necrosis (over 25%) = 4. (ii) Focal and portal inflammation was assessed according to Goodman’s adapted Ishack score [46,47]: none = 0, one focus per 10× objective or less and/or mild inflammation in portal area = 1, two to four foci per 10× objective and/or moderate, some or all portal areas = 2, five to ten foci per 10× objective and/or moderate/marked inflammation in all portal areas = 3, and more than 10 foci per 10× objective and/or marked inflammation in all portal areas = 4. (iii) Histoarchitecture criteria were analyzed according to Goodman’s schematic diagram [46]: no changes = 1; mild interphase hepatitis and parenchymal injury but central vein conserved = 2; moderate interface hepatitis with necroinflammatory foci and moderate parenchymal injury and more than 50% of the central vein integrity = 3; and marked interface hepatitis, stromal collapse with indicators of parenchymal injury and loss of more than 50% of the central vein integrity = 4. Steatosis was evaluated as a percentage of lipid droplets, where 1 <5%, 2 <30%, 3 <50% and 4 >50%. The histological examination was carried out using a Leica DM500 microscope (Leica Microsystems) with a high-definition digital camera, Leica ICC50W (Leica Microsystems), connected to LAS EZ software (Leica Application Suite version 3.4.0., Heerbrugg, Switzerland).

### 4.6. In Silico Analysis of Metabolism and Toxicity

Parameters of the metabolism and toxicity of the diterpenoids identified in the metabolomics analysis were evaluated using the ADMETLAB 2.0 (https://admetmesh.scbdd.com/ accessed on 2, 3, 5, 6 and 7 of January 2014) online platform, where SMILES strings were employed throughout the generation process [48].

### 4.7. Statistical Analyses

Data are presented as the average ± standard deviation (SD). A one way analysis of variance (ANOVA) with multiple comparisons and Tukey’s post hoc test were used. A value of *p* < 0.05 was considered significant. Analyses were performed using Graphpad Prism 9.1.0. (Boston, MA, USA).

## 5. Conclusions

The results of the present study showed that the mice fed with a high-fat diet that received a two-week treatment with organic extracts (methanol, dichloromethane and petroleum ether) from *Azorella compacta* had neither reduced hyperglycemia nor body weight. Furthermore, our findings indicate that these fractions exacerbated the liver alterations in the HFD-fed mice, such as the serum levels of hepatic enzymes. These findings suggest that high doses of *A. compacta* products may induce liver damage in patients with metabolic diseases.

## Figures and Tables

**Figure 1 pharmaceuticals-17-00746-f001:**
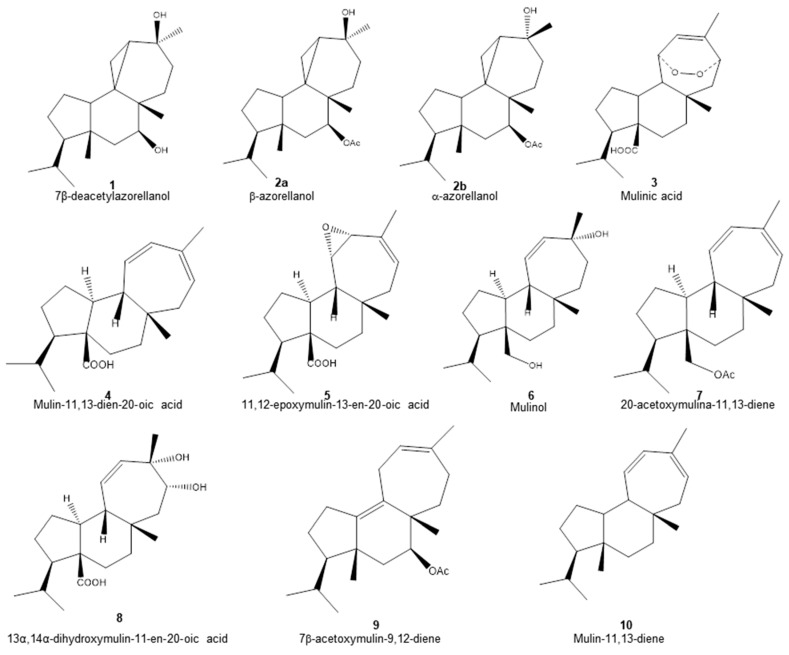
Chemical structure of diterpenoids identified in the organic fractions obtained from *Azorella compacta* by UHPLC-MS.

**Figure 2 pharmaceuticals-17-00746-f002:**
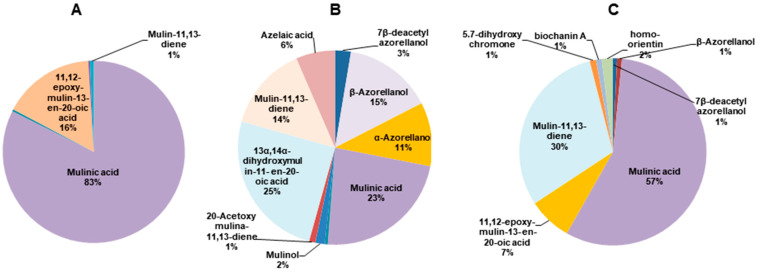
Pie charts of the relative abundance of diterpenoids identified by UHPLC-MS in organic fractions obtained from *Azorella compacta*. (**A**) Petroleum ether fraction; (**B**) dichloromethane fraction and (**C**) methanol fraction from *Azorella compacta*. Compounds with less than 1% abundance in each fraction are not shown.

**Figure 3 pharmaceuticals-17-00746-f003:**
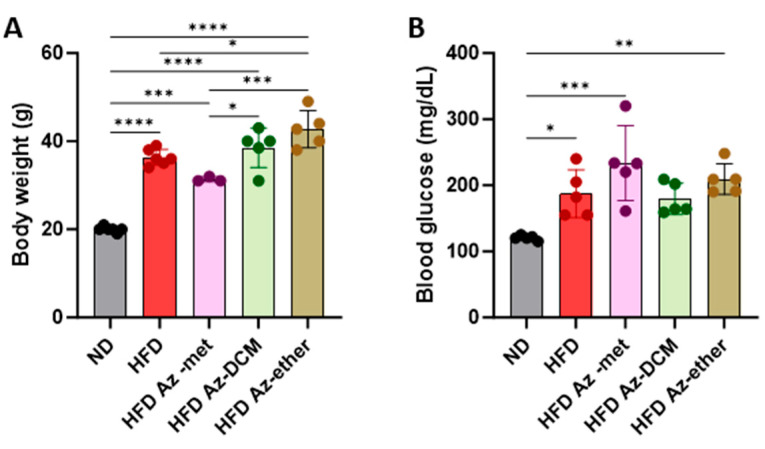
Body weight (**A**) and blood glucose (**B**) levels in animals treated with *Azorella compacta* organic extracts. ND, normal diet; HFD, high-fat diet; HFD Az-met, HFD-fed mice treated with the methanol fraction of *A. compacta*; HFD Az-DCM, HFD-fed mice treated with the dichloromethane fraction of *A. compacta*; and HFD Az-ether, HFD-fed mice treated with the petroleum ether fraction of *A. compacta.* * *p* < 0. 05; ** *p* < 0.005; *** *p* < 0.0005; **** *p* < 0.0001; *n* = 5 each group.

**Figure 4 pharmaceuticals-17-00746-f004:**
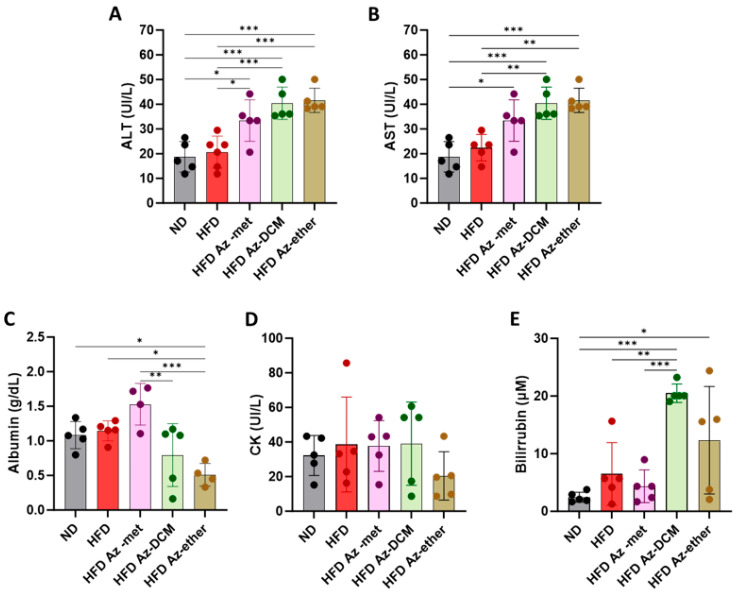
Biochemical parameters of hepatic damage in the serum of mice treated with *Azorella compacta* (*A. compacta*) organic extracts. (**A**,**B**) Serum levels of hepatic enzymes: ALT, alanine aminotransferase (**A**); AST, aspartate aminotransferase (**B**). (**C**) Serum levels of albumin. (**D**) Serum levels of CK, creatine kinase. (**E**) Serum levels of bilirubin. ND, normal diet; HFD, high-fat diet; HFD Az-met, HFD-fed mice treated with the methanol fraction of *A. compacta*; HFD Az-DCM, HFD-fed mice treated with the dichloromethane fraction of *A. compacta*; and HFD Az-ether, HFD-fed mice treated with the petroleum ether fraction of *A. compacta.* * *p* < 0.05; ** *p* < 0.001; *** *p* < 0.0005; *n* = 5 each group.

**Figure 5 pharmaceuticals-17-00746-f005:**
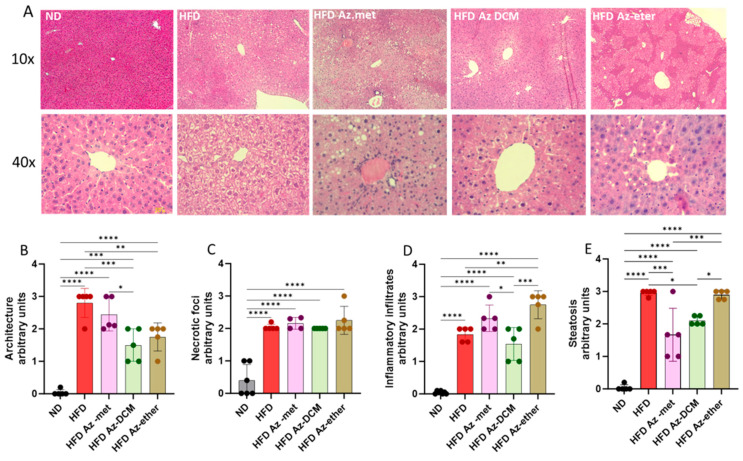
Histopathological analysis of liver sections of animals treated with organic fractions from *Azorella compacta* (*A. compacta*). (**A**) Representative images of hematoxylin/eosin staining of liver sections from animals treated with ND, normal diet; HFD, high-fat diet; HFD Az-met, HFD-fed mice treated with the methanol fraction of *A. compacta*; HFD Az-DCM, HFD-fed mice treated with the dichloromethane fraction of *A. compacta*; and HFD Az-ether, HFD-fed mice treated with the petroleum ether fraction of *A. compacta*. (**B**) Analysis of tissue architecture. (**C**) Analysis of necrotic foci. (**D**) Analysis of inflammatory infiltration. (**E**) Analysis of steatosis. * *p* < 0.05; ** *p* < 0.001; *** *p* < 0.0005; **** *p* < 0.0001; *n* = 5 each group.

**Table 1 pharmaceuticals-17-00746-t001:** Identification of compounds present in organic extracts of *Azorella compacta* by UHLPC/MS.

Molecule	Name	RT (min)	Mass *m*/*z*	Petroleum Ether Fraction (s. Intensity)	DCM Fraction (s. Intensity)	MeOH Fraction (s. Intensity)
**1**	7β-deacetylazorellanol (azorellan-13α,7β-diol)	6.03	306.2	0	2790	1518
**2a**	β-azorellanol (azorellan-13β-hydroxy-7β-yl acetate)	6.15	348.2	0	15,769	1811
**2b**	α-azorellanol (azorellan-13α-hydroxy-7β-yl acetate)	5.45	348.2	0	11,261	0
**3**	Mulinic acid (mulin-11,14-peroxi-12-en-20-oic acid)	8.49	334.2	276,146	24,558	121,212
**4**	Mulin-11,13-dien-20-oic acid	10.41	302.2	1162	493	0
**5**	11,12-epoxymulin-13-en-20-oic acid (mulin-11,12-epoxy-13-en-20-oic acid)	7.7	318.2	54,656	0	16,065
**6**	Mulinol (mulin-11-en-13α,20-diol)	8.3	306.2	0	1694	0
**7**	20-acetoxymulina-11,13-diene (20-hydroxymulin-11,13-dienyl acetate)	7.83	330.2	0	1121	0
**8**	13α,14α-dihydroxymulin-11-en-20-oic acid (mulin-13α,14α-dihydroxy-11-ene-20-oic acid)	5.38	336.2	0	26,551	0
**9**	7β-acetoxymulin-9,12-diene (mulin-9,12-dien-7β-yl acetate)	7.6	330.2	1099	0	0
**10**	Mulin-11,13-diene	10.58	272.2	2198	15,021	64,755
**11**	5,7-dihydroxychromone	2.7	178.0	0	0	2361
**12**	Biochanin A	8.86	284.2	0	0	2018
**13**	Azelaic acid	2.42	188.1	0	6919	0
**14**	Homoorientin	0.71	448.1	0	0	4113

RT: retention time; s. intensity: signal intensity; DCM: dichloromethane; MeOH: methanol.

**Table 2 pharmaceuticals-17-00746-t002:** Hepatic toxicity and metabolism parameters obtained in silico for the diterpenoids identified from the organic fractions of *Azorella compacta*.

Molecule	H-HT (*p*)	DILI (*p*)	CYP1A2 Substrate	CYP2C19 Substrate	CYP2D6 Inhibitor	CYP2D6 Substrate	CYP2C9 Substrate	CYP3A4 Inhibitor	CYP3A4 Substrate	SR-MMP	SR-p53
**1**	0.371	0.018	+	+++	---	---	---	+	+	+++	+
**2a,b**	0.224	0.124	--	+++	--	---	---	-	+	++	+++
**3**	0.360	0.050	++	+++	---	--	+	--	-	+++	++
**4**	0.614	0.011	-	+++	---	-	++	-	+	++	+
**5**	0.263	0.027	+	+++	---	--	-	---	--	++	+
**6**	0.210	0.035	-	++	---	---	---	++	++	+++	---
**7**	0.417	0.086	--	+++	-	--	---	++	++	+	+
**8**	0.246	0.015	+	++	---	---	-	-	-	+++	+
**9**	0.173	0.798	--	+++	---	--	-	+	+	--	+
**10**	0.466	0.023	+	+++	+	++	--	++	++	+	---

H-HT, human hepatotoxicity; DILI, drug-induced liver injury; *p*, probability of being toxic, with 0 being no risk and 1 being high risk; SR-MMP, stress response-induced mitochondrial membrane potential; SR-p53, stress response of p53. The symbols represent the probability values range: 0–0.1 (---), 0.1–0.3 (--), 0.3–0.5 (-), 0.5–0.7 (+), 0.7–0.9 (++), and 0.9–1.0 (+++).

## Data Availability

The raw data supporting the conclusions of this article will be made available by the authors on request.

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
