# Peer review of "Azorella compacta Organic Extracts Exacerbate Metabolic Dysfunction-Associated Fatty Liver Disease in Mice Fed a High-Fat Diet"

_pharmaceuticals, 2024, doi:10.3390/ph17060746_

Round 1

Reviewer 1 Report

Comments and Suggestions for Authors

The study, titled "Azorella compacta Organic Extracts Exacerbate Metabolic Dysfunction-Associated Fatty Liver Disease (MAFLD) in Mice Fed a High-Fat Diet," examines the effects of A. compacta, a plant known for its high diterpenoid content, on liver function. This research is particularly important as A. compacta infusions have been used in the past to treat diabetes and inflammatory diseases. Previous studies have demonstrated a blood glucose-lowering effect of certain diterpenoids in rats, suggesting a potential therapeutic benefit.

However, the study investigates the possibility that preparations of A. compacta, which are likely to contain various terpenoids, could cause hepatotoxicity, particularly in patients with pre-existing metabolic diseases such as diabetes and obesity. One of the alarming findings was an increase in total bilirubin levels in the subjects, indicating possible liver damage.

In light of these findings, it is crucial to conduct a thorough dose-response evaluation to assess the liver toxicity of the organic extracts of A. compacta. This will help determine safe dosages and potentially mitigate the risk of hepatotoxic effects in future therapeutic use, particularly in vulnerable populations with underlying metabolic disorders.

This summary should be revised as soon as the text presents ideas that are unclear.

The paper should be revised to be more factual and not speculative.

Reviewer 2 Report

Comments and Suggestions for Authors

Azorella compacta organic extracts exacerbate metabolic dysfunction-associated fatty liver disease (MAFLD) in mice fed a high fat diet, finds possible liver toxicity in organic extracts of a traditionally used herb when given to rats over a 2-week time span.  This instead of relieving complications from a high fat diet.  The possible disconnect between the study and traditional use of the plant extract is the use of organic extraction with further fractionation of the extract into 3-parts with somewhat different compositions.  This is noted by the authors.  In general, the parameters measured seem appropriate for the study goals.  It is not noted that the statistics account for multiple samples, and this is potentially a problem for the glucose comparison in Fig 3B between, ND and HFD samples and in a few other cases where the p-value is near 0.05. There are a few typos in the text-

Line 78 (e)stablished

Line 190-191 “was” dominated “by”

Figure 2 Comp(o)unds

Line 220 (Figure  (2) 4)

Comments on the Quality of English Language

Needs a little work to meet journal standards.

Reviewer 3 Report

Comments and Suggestions for Authors

Dear Editor and Authors,

The manuscript ‘Azorella compacta organic extracts exacerbate metabolic dysfunction-associated fatty liver disease (MAFLD) in mice fed a  high fat diet’ by Jessica Zúñiga-Hernandez1, Matías Quiñones San Martin1,2, Benjamín Figueroa1, Ulises Novoa1, Francisco 5 Monsalve3, Mitchell Bacho4, Aurelio San-Martin5 and Daniel R. González is a research study on Azorella compacta extract influence on liver health in model research on mice. Although, the plant infusion are used in traditional medicine to cure inflammatory diseases, diabetes, renal and hepatic conditions the Authors found that the organic solvent extract are rather toxic.

Latin names should be in italics

Line 26 extracts were rich in

Line 97 As the results of the research are rather negative I would rather use solvents that are used in food such as water, ethanol or hexane. Maybe there were some solvent residues in the extracts even after they were dried.

line 105 give gradient details and separation duration. More details on mass spectrometry parameters are also needed. What standards were used?

Only relative abundance is given while content should be given. As nothing is a toxin and everything is. All depends on a dose.

Line 112 Number of ethic commission number permission should be given.

How many animals were in a group?

Figure 1. Give the names of the compounds in a caption

Line 300 it should be ‘rich’ not ‘enriched’ as enriched means something was added

Line 357 Here you have a short discussion that Azorella is traditionally extracted with water only. Maybe the organic solvents were the problem?

48 references is ok

Cordially

Round 2

Reviewer 1 Report

Comments and Suggestions for Authors

The authors have responded positive.

Reviewer 3 Report

Comments and Suggestions for Authors

Dear Editor and Authors,

The manuscript is improved after the revision

Cordially